# SCMF: Lightweight Retrieval-Augmented Generation via Retrieval Vector Compression

## Abstract

With the widespread adoption of Retrieval-Augmented Generation (RAG) in knowledge-intensive tasks, efficiency bottlenecks become increasingly evident: storing and retrieving large-scale high-dimensional embeddings incur substantial storage and computation costs. To address this challenge, we propose the Semantic Compressed Memory Framework (SCMF), a lightweight and traceable indexing paradigm tailored for large-scale RAG. SCMF first projects document embeddings into a low-dimensional semantic space, and then discretizes them into compact Semantic Memory Units (SMUs) via Residual Vector Quantization (RVQ). Each SMU is explicitly linked to its corresponding Raw Knowledge Unit (RKU) through a semantic inverted index, which enables efficient CRUD operations while preserving the traceability of retrieval results. During retrieval, SCMF performs Approximate Nearest Neighbor (ANN) search in the SMU space, followed by a two-stage re-ranking strategy that combines sparse retrieval (BM25) and dense retrieval, thereby achieving efficient and accurate evidence localization. Experimental results demonstrate that SCMF substantially reduces storage costs and retrieval latency while preserving explicit traceability to the original knowledge units, significantly outperforming mainstream vector indexing methods.

## 1 Introduction

In recent years, Large Language Models (LLMs) have demonstrated strong capabilities in tasks such as question answering, reasoning, and knowledge-driven generation (Vaswani et al., 2017; Brown et al., 2020). However, since their knowledge is fixed within model parameters, LLMs often struggle to handle dynamic knowledge and domain-specific information, and they are prone to hallucinations (Huang et al., 2024). To address this issue, Retrieval-Augmented Generation (RAG) has gradually become a mainstream solution. By introducing an external retrieval corpus and retrieving relevant documents during inference, RAG effectively mitigates hallucinations and knowledge obsolescence, and it has been widely applied in knowledge-intensive tasks (Lewis et al., 2020; Shi et al., 2023; Izacard et al., 2023).

However, as the size of retrieval corpora continues to grow, traditional RAG frameworks face bottlenecks of high storage cost and large retrieval latency (Borgeaud et al., 2022). Some studies attempt to introduce more intelligent optimization strategies into the retrieval–generation interaction. For example, Self-RAG (Asai et al., 2024) dynamically adjusts the retrieval and generation pipeline through self-reflection and critique, significantly improving response quality in complex query scenarios. Other methods incorporate external structured knowledge. GraphRAG (Han et al., 2024), for instance, constructs local "problem–paragraph" knowledge graphs to enhance reasoning ability. Although these approaches achieve notable improvements in retrieval quality and reasoning capability, their core optimizations remain focused on retrieval algorithms or knowledge fusion strategies, without systematic consideration of the underlying indexing structures.

At the indexing level, existing vector databases typically rely on IVF-Flat (Baranchuk et al., 2018), IVF-PQ/OPQ (Jegou et al., 2010), or HNSW (Malkov & Yashunin, 2018), which achieve good performance on the latency–recall trade-off (Mohoney et al., 2024). Nevertheless, these methods require storing additional codes for each vector, their cluster partitions lack semantic interpretability, and their dynamic updates incur high costs. These limitations restrict their applicability in large-scale and continuously evolving RAG scenarios (Zhong et al., 2025; Han et al., 2023).

To address this challenge, we propose the *Semantic Compressed Memory Framework* (SCMF) from the perspective of vector storage optimization for retrieval corpora. Inspired by vector-quantized autoencoders (VQ-VAE) (Van Den Oord et al., 2017), SCMF reduces storage and retrieval overhead while preserving semantic expressiveness and knowledge traceability, thereby supporting efficient retrieval. Unlike traditional approaches that directly rely on high-dimensional embeddings, SCMF first encodes raw documents into continuous representations and then compresses them into discrete *Semantic Memory Units* (SMUs) via latent space mapping (PCA) and Residual Vector Quantization (RVQ). These SMUs are stored in a dynamically maintainable *Semantic Memory Bank* (SMB) and linked to the original *Raw Knowledge Units* (RKUs) through inverted indices, ensuring the traceability of retrieval results. During retrieval, SCMF performs efficient Approximate Nearest Neighbor (ANN) search in the SMU space, followed by a two-stage re-ranking strategy that combines sparse retrieval (BM25) and dense retrieval, thus achieving controllable storage cost and reduced retrieval latency while preserving evidence completeness and accuracy. Moreover, thanks to the integration of SMB and inverted indexing, SCMF natively supports efficient CRUD operations (Create, Read, Update, Delete), enabling dynamic maintenance without the need to rebuild large-scale indexes. This makes it particularly well-suited for large-scale and continuously evolving knowledge bases. controllable storage cost and reduced retrieval latency while preserving evidence completeness and accuracy. Our main contributions are summarized as follows:

- We propose SCMF, a novel indexing framework for large-scale and continuously updated RAG, which reduces storage cost while preserving knowledge traceability.

- SCMF integrates PCA and two-level RVQ to compress dense embeddings into discrete SMUs, achieving nearly two orders of magnitude storage reduction and enabling efficient matching with semantic consistency.

- Extensive experiments on QA benchmarks demonstrate that SCMF significantly lowers storage and retrieval latency, providing an efficient and interpretable solution for practical RAG systems.

## 2 RELATED WORK

### 2.1 RETRIEVAL OPTIMIZATION IN RAG FRAMEWORKS

Retrieval-Augmented Generation (RAG) combines external retrieval with generative models to enhance factuality and reduce hallucinations in knowledge-intensive tasks (Guu et al., 2020; Lewis et al., 2020; Karpukhin et al., 2020). A typical example is Dense Passage Retriever (DPR), which matches high-dimensional embeddings over large corpora and couples them with generators such as T5 or BART. This design enables end-to-end retrieval–generation pipelines, laying the foundation for open-domain QA and reasoning by allowing models to ground their outputs in external evidence (Izacard & Grave, 2020; Gao et al., 2023).

Subsequent work improves retrieval strategies and fusion mechanisms to overcome RAG's bottlenecks (Fan et al., 2024; Zhao et al., 2024). Fusion-in-Decoder (FiD) (Izacard & Grave, 2020) integrates multiple retrieved documents during decoding, improving robustness. Self-RAG (Asai et al., 2024) incorporates self-reflection to revise retrieved evidence, while GraphRAG (Han et al., 2024) leverages knowledge graphs for multi-hop reasoning and richer relation modeling.

More recent studies extend RAG with new designs. RAG+ (Wang et al., 2025) uses a dual corpus of knowledge and examples for joint retrieval. Vendi-RAG (Rezaei & Dieng, 2025) introduces a semantic diversity score to balance relevance and diversity in multi-hop retrieval. HeteRAG (Yang et al., 2025) separates representations for retrieval and generation to improve efficiency and semantic completeness. LevelRAG (Zhang et al., 2025) decomposes queries into sub-queries, combining sparse, dense, and graph-based retrievers for hybrid retrieval.

Although these approaches enhance retrieval and fusion, they focus mainly on document utilization. From a deployment perspective, they still suffer from efficiency bottlenecks, as large-scale high-dimensional embeddings increase storage and latency, and most methods lack explicit mechanisms for knowledge traceability.

## 2.2 EFFICIENT REPRESENTATION AND STORAGE OPTIMIZATION IN RAG

With the rapid expansion of retrieval corpora, RAG systems face increasingly prominent challenges in storage and retrieval efficiency. Traditional vector retrieval often relies on high-dimensional embeddings (e.g., 768 dimensions or higher), which cause storage to grow linearly with corpus size and place heavy computational burdens on Approximate Nearest Neighbor (ANN) search. Improving efficiency while preserving semantic fidelity has thus become a key research focus.

Among existing approaches, vector quantization (VQ) is one of the most widely adopted solutions. Product Quantization (PQ) (Jegou et al., 2010) and Residual Vector Quantization (RVQ) (Chen et al., 2010) discretize high-dimensional embeddings into compact codebooks, reducing storage and accelerating similarity computation (Babenko & Lempitsky, 2014). Beyond VQ, researchers explore knowledge compression and low-rank approximation. Distillation methods compress the knowledge of large models into smaller models or latent spaces (Hinton et al., 2015; Zhang et al., 2023), while parameter-efficient techniques such as low-rank decomposition (Houlsby et al., 2019; Hu et al., 2022) reduce computational overhead and memory usage. In parallel, external or parameterized memory mechanisms have been introduced. For instance, kNN-LM (Khandelwal et al., 2019) augments generation with nearest-neighbor search over an external memory, and RETRO (Borgeaud et al., 2022) integrates large-scale retrieval with language models to enhance knowledge coverage.

Overall, these methods demonstrate encouraging progress, yet two limitations persist: (1) compression inevitably sacrifices semantic fidelity, lowering the accuracy and reliability of retrieved evidence; and (2) most lack explicit backtracking to the original knowledge, limiting interpretability in reasoning tasks. To address these issues, our work introduces a semantic compressed memory perspective, mapping raw knowledge into discrete semantic units while preserving explicit links to the original text, thereby achieving both efficiency and traceability.

## 3 METHOD

In this section, we present the SCMF in detail. Specifically, we first describe the core components of the framework, then explain the compression mechanism and its training objectives. We next outline the end-to-end workflow for index construction and retrieval, and introduce the CRUD operations that enable dynamic memory management. Finally, we summarize the implementation details. For clarity, an overview of the framework is provided in Figure 1.

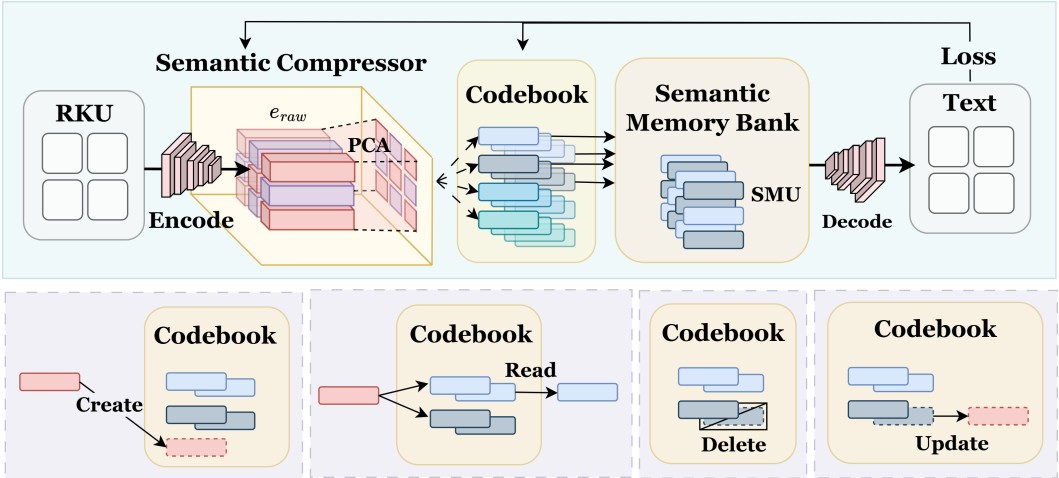

Figure 1: Training process of SCMF. RKUs are encoded, projected by PCA, quantized into codewords, and stored in the Semantic Memory Bank (SMB) with explicit links to the original RKUs. The lower panel illustrates CRUD operations for dynamic SMU/SMB maintenance.

## 3.1 Core Components

**Raw Knowledge Unit (RKU).** The atomic input segment—typically a paragraph or a document chunk of about 600 tokens—is encoded by a fixed text embedding model, denoted as $\mathrm{DocEncoder}$, into a high-dimensional vector $\mathbf{e} \in \mathbb{R}^d$ with $d{=}3072$.

**Semantic Compressor (PCA).** A fixed PCA projection replaces the learnable encoder $\mathcal{E}$ and maps $\mathbf{e}$ into a latent vector $\mathbf{z} \in \mathbb{R}^{d_\ell}$, with the latent dimension set to $d_\ell = 270$, via $\mathbf{z} = W^\top(\mathbf{e} - \mu)$.

**Semantic Memory Unit (SMU).** A compressed, discrete prototype that captures the core semantics of one or more RKUs. Each SMU is indexed by a pair $(i, j)$ obtained from a two-level residual vector quantization (RVQ) over $\mathbf{z}$. Let $\mathcal{C}_1$ and $\mathcal{C}_2$ be codebooks with entry dimension $d_\ell$; the SMU prototype is

$$\hat{\mathbf{z}} = \mathbf{c}_{1,i} + \mathbf{c}_{2,j}, \quad \mathbf{c}_{1,i} \in \mathcal{C}_1, \ \mathbf{c}_{2,j} \in \mathcal{C}_2 \tag{1}$$

Thus, an SMU is uniquely identified by $(i, j)$ or, equivalently, by its prototype $\hat{\mathbf{z}}$.

**Semantic Memory Bank (SMB).** The SMB stores the mappings between RKUs, their quantized indices $(i, j)$, and the corresponding prototypes $\hat{\mathbf{z}}$, serving as the central repository for SCMF. Let $\mathcal{P} = \{\hat{\mathbf{z}}_{i,j}\}$ denote the set of all SMU prototypes stored in the SMB.

**Semantic Retriever.** Given a query embedding $\mathbf{z}_q$, the retriever performs approximate nearest neighbor (ANN) search over $\mathcal{P}$ using cosine similarity to obtain Top-$K$ candidates ($K{=}16$), followed by re-ranking with semantic similarity and metadata such as frequency, recency, and credibility.

**Terminology clarification.** We distinguish between the Semantic Memory Unit (SMU), which denotes a single quantized prototype indexed by $(i, j)$, and the Semantic Memory Bank (SMB), which denotes the entire repository that stores the set of SMUs and their mappings to the corresponding Raw Knowledge Units (RKUs). In the remainder of this paper, "SMU space" refers to the collection of all SMU prototypes contained in the SMB.

## 3.2 Semantic Compression and Training Objectives

**PCA projection.** To reduce the dimensionality of the raw DocEncoder embeddings $\mathbf{e} \in \mathbb{R}^d$, where $d = 3072$, we first apply Principal Component Analysis (PCA) as a fixed linear projection:

$$\mathbf{z} = W^\top(\mathbf{e} - \mu), \qquad W \in \mathbb{R}^{d \times d_\ell} \tag{2}$$

Here $W$ and $\mu$ denote the PCA projection matrix and mean vector, respectively, and the latent dimension is set to $d_\ell = 270$. This step serves two purposes: (i) it decorrelates the embedding dimensions and concentrates most of the variance into a lower-dimensional latent subspace, and (ii) it provides a compact representation that reduces both memory footprint and subsequent quantization error. Unlike trainable encoders, the PCA projection is fixed and lightweight, ensuring stable preprocessing while avoiding additional training overhead. We specifically choose PCA because our goal is not to learn new semantic features, but to retain the major variance of the original Raw Knowledge Units (RKUs) as compact surrogates that can be efficiently aligned with the quantization codebooks. This design preserves the essential semantic information while minimizing training cost, making PCA a simple yet effective preprocessing step for large-scale semantic memory compression.

**Two-level RVQ.** After PCA projection, we discretize the latent vector $\mathbf{z}$ using a two-level Residual Vector Quantization (RVQ):

$$\mathbf{c}_1 = \arg\min_{\mathbf{c} \in \mathcal{C}_1} \|\mathbf{z} - \mathbf{c}\|_2, \quad \mathbf{r}_1 = \mathbf{z} - \mathbf{c}_1, \quad \mathbf{c}_2 = \arg\min_{\mathbf{c} \in \mathcal{C}_2} \|\mathbf{r}_1 - \mathbf{c}\|_2, \quad \hat{\mathbf{z}} = \mathbf{c}_1 + \mathbf{c}_2 \tag{3}$$

Here, $\mathcal{C}_1$ and $\mathcal{C}_2$ are codebooks of size $M$, where $M = 32\mathrm{k}$. The two-level structure allows the system to progressively refine the approximation of $\mathbf{z}$. The first quantizer $\mathcal{C}_1$ captures coarse semantic structure, while the second quantizer $\mathcal{C}_2$ encodes residual information to improve fidelity. This hierarchical design strikes a balance between compression ratio and reconstruction accuracy, enabling the system to map continuous vectors into discrete prototypes that are both compact and semantically representative.

**Codebook objective.** Since PCA $(W, \mu)$ is fixed and no neural decoder is trained, we learn the codebooks $\mathcal{C}_1, \mathcal{C}_2$ by minimizing the quantization distortion of latent vectors:

$$\mathcal{L}_{\mathrm{vq}} = \left\|\mathbf{z} - \hat{\mathbf{z}}\right\|_2^2, \qquad \hat{\mathbf{z}} = \mathbf{c}_{1,i} + \mathbf{c}_{2,j}, \ \ \mathbf{c}_{1,i} \in \mathcal{C}_1, \ \mathbf{c}_{2,j} \in \mathcal{C}_2 \tag{4}$$

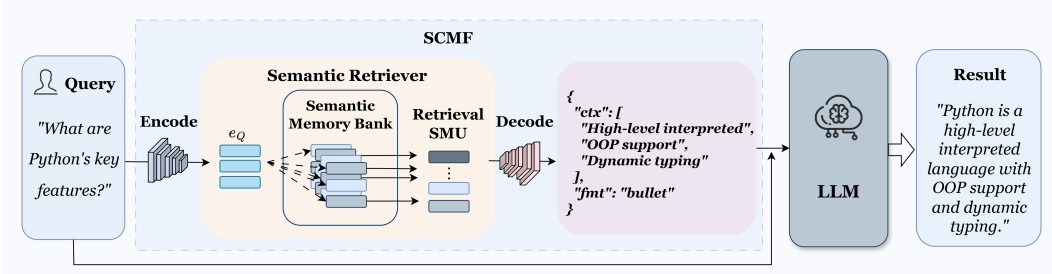

Figure 2: Inference process of SCMF. The query is encoded and matched with SMU prototypes via ANN in the latent space, enabling fast retrieval. The linked RKUs are then retrieved in their full original form from the SMB, assembled as context, and passed to the LLM to generate the final response.

In practice, we fit $\mathcal{C}_1, \mathcal{C}_2$ via residual k-means or EMA-style updates over assignments. Optionally, a commitment-style term can stabilize assignments without training an encoder:

$$\mathcal{L} = \mathcal{L}_{\mathrm{vq}} + \beta \left\| \mathrm{sg}[\hat{\mathbf{z}}] - \mathbf{z} \right\|_2^2 \tag{5}$$

where $\mathrm{sg}[\cdot]$ stops gradients into the prototypes; $\beta$ is a small constant, where $\beta = 0.5$.

### 3.3 WORKFLOW: TRAINING AND INFERENCE

**Codebook Pretraining.** We first fit a PCA on a large retrieval corpus to obtain $(W, \mu)$. We then learn the RVQ codebooks $\mathcal{C}_1, \mathcal{C}_2$ over latent vectors $\mathbf{z} = W^\top (\mathbf{e} - \mu)$ using residual k-means clustering to minimize quantization distortion. During this process, we build the SMB by inserting the mappings $x \mapsto (i,j)$ and $(i,j) \mapsto \hat{\mathbf{z}}$ for each RKU $x$. Training proceeds in a mini-batch fashion over the corpus.with pseudocode provided in Appendix 1.

**Inference.** At runtime, SCMF executes the following process: query encoding $\rightarrow$ SMU retrieval $\rightarrow$ evidence localization (BM25 retrieves Top-20 candidates, dense re-ranking selects Top-2 per SMU; the dense retriever shares the same encoder as DocEncoder) $\rightarrow$ context assembly with the full retrieved RKUs ($\leq 512$ tokens) $\rightarrow$ LLM generation. The overall inference workflow is illustrated in Figure 2, and summarized in Appendix 2.

### 3.4 CRUD OPERATIONS AND MAINTENANCE

SCMF natively supports incremental memory management through the standard CRUD operations. For *Create* (C), a new RKU $x$ is first embedded as $\mathbf{e} \in \mathbb{R}^d$ and projected into the latent vector

$$\mathbf{z} = W^\top (\mathbf{e} - \mu) \tag{6}$$

Residual vector quantization (RVQ) maps $\mathbf{z}$ to the index $(i,j)$ with prototype $\hat{\mathbf{z}}$, and the mappings $x \mapsto (i,j)$ and $(i,j) \mapsto \hat{\mathbf{z}}$ are inserted into the SMB. For *Read* (R), a query $q$ is embedded and projected as $\mathbf{z}_q = W^\top (\mathbf{e}_q - \mu)$, after which the system retrieves the Top-$K$ nearest SMUs $\{\hat{\mathbf{z}}_k\}_{k=1}^K$ via ANN search and then looks up the SMB to obtain the associated evidences $E_{\mathrm{ret}}$. For *Update* (U), a modified RKU $x'$ is embedded, projected, and re-quantized to $(i',j')$, replacing the previous mapping of $x$ in the SMB:

$$(x \mapsto (i,j)) \mapsto (x' \mapsto (i',j')) \tag{7}$$

Finally, for *Delete* (D), all links $x \mapsto (i,j)$ associated with obsolete RKUs are removed. If an SMU $(i,j)$ becomes unreferenced, it is marked as inactive and recycled:

$$(i,j) \text{ inactive} \mapsto \text{recycle} \tag{8}$$

This design ensures efficient one-time storage with instant retrieval, eliminating the need to re-encode the entire memory bank.

### 3.5 IMPLEMENTATION NOTES

We adopt the following default settings in SCMF. Each RKU is a 600-token chunk encoded into a 3072-dimensional embedding. This embedding is projected into a 270-dimensional latent space via a fixed PCA transformation $(W, \mu)$, and subsequently quantized using a two-level RVQ with 32k codewords per level (64k in total). During inference, the retriever selects Top-16 SMUs per query. For each SMU, BM25 generates 20 candidate sentences, and dense re-ranking selects the Top-2 as evidence. The retrieved RKUs are then directly assembled into the context with a maximum budget of 512 tokens. We apply deduplication with a similarity threshold of 0.8 to 0.85.

## 4 EXPERIMENT

To evaluate the performance of SCMF, we conduct experiments in two stages: Compression Configuration Selection and Framework Performance Validation. The first stage identifies optimal PCA and RVQ settings by analyzing variance, quantization distortion, ranking consistency, and retrieval latency. The second stage applies these settings to downstream QA tasks, comparing SCMF with PCA-only and other compression baselines, and assessing its generalizability under different reasoning paradigms.

### 4.1 EXPERIMENTAL SETUP

We evaluate SCMF on five QA benchmarks (MultiDocQA (Feng et al., 2021), Hotpot-Long (Yang et al., 2018), NarrativeQA (Kočiský et al., 2018), GovReport (Huang et al., 2021), QASPER (Dasigi et al., 2021)) and use STS-B (Cer et al., 2017) for quantization efficiency analysis. Documents are split into 600-token units, encoded into 3072-dimensional embeddings, reduced to 270 dimensions with PCA, and quantized by two-level RVQ. At inference, the retriever selects Top-16 SMUs, expands candidates with BM25 and dense reranking, and assembles a 512-token context. We report retrieval accuracy (EM, nDCG@10, MRR), efficiency (mean/p95 latency), and quantization metrics ($\Delta$cos, MSE, Spearman $\rho$, J@20). Downstream QA tasks use GPT-4o-mini (Achiam et al., 2023), Qwen-Long, and Qwen-Max (Bai et al., 2023), with Full-Dense, PCA-only, PQ/OPQ (Jegou et al., 2010; Ge et al., 2013), and their IVF variants as baselines. Detailed training settings and evaluation protocols are provided in the Appendix 7.

### 4.2 COMPRESSION CONFIGURATION SELECTION

Table 1: RVQ sweep on STS-B ($d = 270$ PCA). Layers $L$ and codebook sizes $M$ compared by distortion and ranking. Metrics: $\Delta$cos (cosine distance), MSE (mean squared error), Spear. $\rho$ (Spearman correlation vs. PCA), J@20 (top-20 overlap). Best at ($L = 2, M = 32$k).

| $(L, M)$ | Bits | $\Delta$cos | MSE | Spear. $\rho$ | J@20 |
|---|---|---|---|---|---|
| $(1, 8\text{k})$ | 13 | $0.085 \pm 0.010$ | $0.018 \pm 0.004$ | $0.960 \pm 0.010$ | $0.86 \pm 0.03$ |
| $(1, 16\text{k})$ | 14 | $0.070 \pm 0.008$ | $0.014 \pm 0.003$ | $0.965 \pm 0.008$ | $0.89 \pm 0.03$ |
| $(2, 16\text{k})$ | 28 | $0.030 \pm 0.006$ | $0.0050 \pm 0.0015$ | $0.985 \pm 0.006$ | $0.93 \pm 0.02$ |
| $(2, 32\text{k})$ | 30 | $0.025 \pm 0.005$ | $0.0038 \pm 0.0012$ | $0.988 \pm 0.005$ | $0.95 \pm 0.02$ |
| $(2, 64\text{k})$ | 32 | $0.023 \pm 0.004$ | $0.0032 \pm 0.0010$ | $0.989 \pm 0.004$ | $0.955 \pm 0.02$ |
| $(3, 32\text{k})$ | 45 | $0.020 \pm 0.004$ | $0.0026 \pm 0.0008$ | $0.991 \pm 0.004$ | $0.960 \pm 0.02$ |

**PCA Variance Analysis for Latent Dimension Selection.** We first analyze the cumulative variance explained by PCA to determine the latent dimension $d$. Figure 3 reports the number of components required to preserve different variance thresholds.

While prior work chose $d$=110 mainly to minimize the storage of PCA vectors, our system stores only the RVQ codebooks (and short codes), making index size largely independent of $d$.

From the curve, the marginal gain in variance slows markedly after $\sim$266 components (70% variance) and becomes very small towards 430 components (80% variance). We therefore select $d$=270 as a balanced point that ensures high fidelity while avoiding diminishing returns beyond the 70% elbow.

**Optimal Configuration Selection in RVQ Quantization.** In the previous PCA analysis, we determine $d = 270$ as the latent dimension by analyzing the cumulative variance explained by PCA. Based on this chosen latent dimension, we further evaluate the quantization performance in the RVQ experiments with different configurations. The experimental results, as shown in Table 1 shows that as the number of RVQ layers ($L$) and the codebook size ($M$) increase, the quantization error (measured by $\Delta$cos and MSE) decreases significantly. Specifically, the $(2, 32\mathrm{k})$ configuration achieves very low distortion ($\Delta$cos $= 0.025 \pm 0.005$, MSE $= 0.0038 \pm 0.0012$) and strong ranking performance (Spearman $\rho = 0.988 \pm 0.005$, J@20 $= 0.95 \pm 0.02$). While larger settings such as $(2, 64\mathrm{k})$ or $(3, 32\mathrm{k})$ yield slightly lower distortion and higher correlation, their improvements are marginal com-

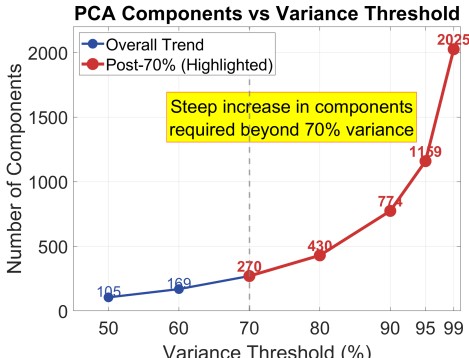

Figure 3: Number of PCA components required to preserve specified levels of variance.

pared to the additional storage and computational cost. Therefore, $(2, 32\mathrm{k})$ is selected as the optimal trade-off, offering near-optimal quantization accuracy and ranking consistency while maintaining practical efficiency.

Further increasing $M$, as in the (2, 64k) and (3, 32k) configurations, continues to reduce quantization errors and improve correlation, but the improvements become marginal while additional storage and computational costs increase. Therefore, the (2, 32k) setting is selected as the optimal configuration, striking the best balance between quantization precision, ranking consistency, and efficiency.

In the subsequent end-to-end retrieval experiments, we evaluate the PCA-only baseline and our RVQ-based retrievers at $d = 270$. As shown in Table 2, the RVQ-based retriever, particularly with $M{=}32\mathrm{k}$, achieves a clear reduction in latency while maintaining comparable retrieval quality. Top-20 coverage remains above 0.93, and the drops in nDCG@10 and MRR are marginal, indicating that SCMF enables efficient large-scale retrieval without compromising semantic fidelity.

Table 2: End-to-end retrieval on STS-B ($d = 270$, $L = 2$), comparing PCA-only and RVQ-based retrievers. Top-20 coverage, candidate size, nDCG@10, relative difference to PCA-only, and MRR are reported as mean $\pm$ variation, while latency is shown as mean / p95 (ms).

| Method | Top-20 Cov. | Cand. (median) | Latency (ms) | nDCG@10 | $\Delta$ vs. PCA | MRR |
|---|---|---|---|---|---|---|
| PCA-only (baseline) | — | — | $75 \pm 15$ | $0.888 \pm 0.007$ | — | $0.810 \pm 0.010$ |
| Ours (M=32k, main) | $0.95 \pm 0.02$ | $280 \pm 40$ | $16 \pm 4$ | $0.880 \pm 0.005$ | $-1.0 \pm 0.5$ p.p. | $0.803 \pm 0.010$ |
| Ours (M=16k, ctrl) | $0.93 \pm 0.03$ | $300 \pm 50$ | $19 \pm 5$ | $0.875 \pm 0.008$ | $-1.5 \pm 0.5$ p.p. | $0.800 \pm 0.010$ |

In conclusion, the latent dimension $d = 270$, chosen based on the PCA analysis, provides a solid foundation for the RVQ-based retriever. In the subsequent RVQ experiments, we ultimately select the combination of $M = 32k$ and $L = 2$. This combination strikes the best balance between quantization precision, ranking consistency, and computational efficiency, making it suitable for subsequent large-scale multimodal tasks and retrieval applications.

### 4.3 SYSTEM-LEVEL EFFICIENCY COMPARISON

To complement retrieval accuracy, we evaluate system-level efficiency across baselines. As shown in Table 3, different indexing strategies present clear trade-offs among accuracy, latency, and storage. Full-Dense achieves the highest accuracy but suffers from high latency and linearly growing storage. PCA-only reduces both but remains corpus-dependent. PQ/OPQ and IVF-based variants yield compact indices, yet aggressive quantization causes accuracy degradation.

In contrast, SCMF attains accuracy close to PCA-only while achieving much lower latency and higher throughput. This gain comes from searching over compact semantic prototypes rather than high-dimensional embeddings, cutting down nearest neighbor search overhead. Notably, SCMF's index size stays nearly constant across corpus scales, since most storage lies in global RVQ codebooks

Table 3: System-level efficiency comparison across indexing methods. We report retrieval quality (nDCG@10 with $\Delta$ vs. Full-Dense in percentage points), mean latency, throughput (QPS), and index size under different corpus scales. SCMF delivers up to $\sim 15\times$ lower latency and $\sim 10\times$ higher throughput than Full-Dense, while preserving comparable retrieval quality and achieving large storage savings versus dense baselines.

| Method | nDCG@10 ( $\Delta$ vs Full ) | Latency (ms) | QPS | Index Size | | |
|---|---|---|---|---|---|---|
| | | | | **10k Docs** | **1M Docs** | **10M Docs** |
| Full-Dense | 0.880–0.895 (—) | 180–260 | 4–5 | 117.2 MB | 11.4 GB | 114.4 GB |
| PCA-only | 0.878–0.892 ($-0.2\sim0.3$ p.p.) | 60–90 | 12–15 | 9.3 MB | 0.9 GB | 9.3 GB |
| PQ-256-8 | 0.865–0.880 ($-1.5\sim2.0$ p.p.) | 30–50 | 20–30 | **0.63 MB** | 63.0 MB | 0.63 GB |
| OPQ-256-8 | 0.872–0.885 ($-0.8\sim1.5$ p.p.) | 35–55 | 18–25 | **0.63 MB** | 63.0 MB | 0.63 GB |
| IVF-PQ-256-8 | 0.859–0.874 ($-2.1\sim2.6$ p.p.) | 15–30 | 35–45 | 0.68 MB | 68.0 MB | 0.68 GB |
| IVF-OPQ-256-8 | 0.867–0.880 ($-1.3\sim2.8$ p.p.) | 18–35 | 30–40 | 0.68 MB | 68.0 MB | 0.68 GB |
| **SCMF (ours)** | 0.878–0.888 ($-0.5\sim1.5$ p.p.) | **12–20** | **50–60** | 70.7 MB | 73.9 MB | **104.0 MB** |

and per-document storage is only short discrete codes. Even with 10M documents, SCMF requires about 104 MB, ensuring scalability and long-term efficiency. Overall, SCMF's semantic compression design preserves retrieval fidelity while drastically lowering system-level costs.

## 4.4 DOWNSTREAM QA VALIDATION

Table 4: Performance comparison on MultiDocQA, Hotpot-Long, and NarrativeQA between PCA-only and SCMF (averaged over 5 random seeds). Results are reported as mean $\pm$ std. Latency is reported as mean / p95 (ms).

| Model | Dataset | PCA-only (EM %) | SCMF (EM %) | $\Delta$ (p.p.) | Latency (ms, PCA) | Latency (ms, SCMF) |
|---|---|---|---|---|---|---|
| GPT-4o-mini | MultiDocQA | $63.2 \pm 0.6$ | $62.6 \pm 0.5$ | $-0.6 \pm 0.2$ | 220 / 310 | 75 / 120 |
| | Hotpot-Long | $59.0 \pm 0.5$ | $58.4 \pm 0.6$ | $-0.6 \pm 0.2$ | 200 / 280 | 68 / 115 |
| | NarrativeQA | $46.1 \pm 0.6$ | $45.6 \pm 0.5$ | $-0.5 \pm 0.2$ | 190 / 260 | 65 / 105 |
| Qwen-Long | MultiDocQA | $64.3 \pm 0.5$ | $63.6 \pm 0.5$ | $-0.7 \pm 0.2$ | 230 / 320 | 85 / 130 |
| | Hotpot-Long | $60.3 \pm 0.6$ | $59.6 \pm 0.5$ | $-0.7 \pm 0.2$ | 215 / 300 | 80 / 125 |
| | NarrativeQA | $46.8 \pm 0.4$ | $46.2 \pm 0.5$ | $-0.6 \pm 0.2$ | 210 / 290 | 77 / 118 |
| Qwen-Max | MultiDocQA | $61.0 \pm 0.6$ | $60.3 \pm 0.5$ | $-0.7 \pm 0.2$ | 240 / 330 | 90 / 135 |
| | Hotpot-Long | $57.0 \pm 0.5$ | $56.2 \pm 0.6$ | $-0.8 \pm 0.2$ | 225 / 310 | 85 / 128 |
| | NarrativeQA | $44.0 \pm 0.5$ | $43.4 \pm 0.5$ | $-0.6 \pm 0.2$ | 215 / 300 | 82 / 120 |

**Evaluation of Accuracy–Latency Trade-off in SCMF.** Table 4 shows that our RVQ-based method substantially reduces retrieval latency across all models. For example, on MultiDocQA with GPT-4o-mini, latency drops from 220 ms (p95: 310 ms) with PCA-only to 75 ms (p95: 120 ms). These gains come with only a marginal accuracy drop, making SCMF effective for scenarios requiring rapid responses. Compared with PCA-only, SCMF incurs a small decrease in EM of $-0.5$ to $-0.8$ percentage points, which remains within an acceptable range. In particular, on Qwen-Long and Qwen-Max, the EM differences are limited to 0.6–0.8 p.p., indicating that the RVQ-based method preserves stability while substantially accelerating inference. In summary, SCMF emphasizes efficiency over a minor accuracy drop, offering a practical solution for large-scale retrieval tasks, especially in applications that demand low latency and high scalability.

**Comparative Evaluation of SCMF and Compression Baselines.** Figure 4 compares SCMF with common compression baselines across five datasets (MultiDocQA, HotpotQA-Long, NarrativeQA, GovReport, QASPER) and four models (GPT-4o-mini, Qwen-Max, Gemini 1.5 Flash, Llama-3.1-8B-Instruct). Full-Dense achieves the highest scores, as expected from its uncompressed nature. Among the baselines, PCA-only remains close to Full-Dense, while PQ-256-8 and OPQ-256-8 show drops of around 2–3 points. We also include IVF-PQ and IVF-OPQ for completeness; because these methods apply more aggressive quantization, they introduce greater information loss and therefore consistently underperform PCA-only and SCMF, typically by 2–3 percentage points. In contrast, SCMF stays close to PCA-only, generally within one point of Full-Dense, and in some cases even surpasses it, while consistently outperforming both PQ/OPQ and their IVF variants. These results

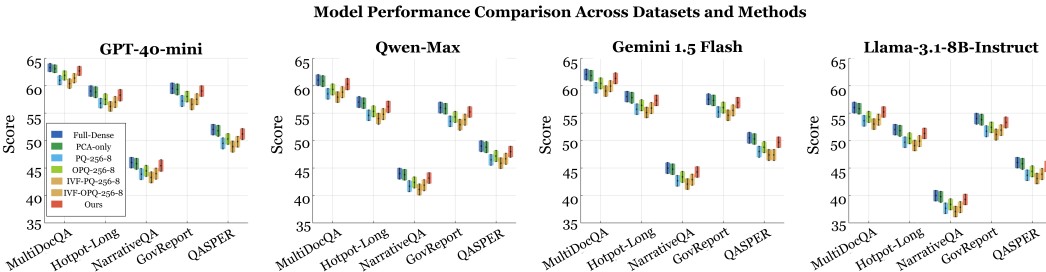

Figure 4: Performance comparison on five QA benchmarks using four different LLMs.

demonstrate that although compression inevitably introduces some loss, SCMF provides a favorable balance between accuracy and efficiency, and this trend holds robustly across different models and datasets, highlighting its applicability and robustness.

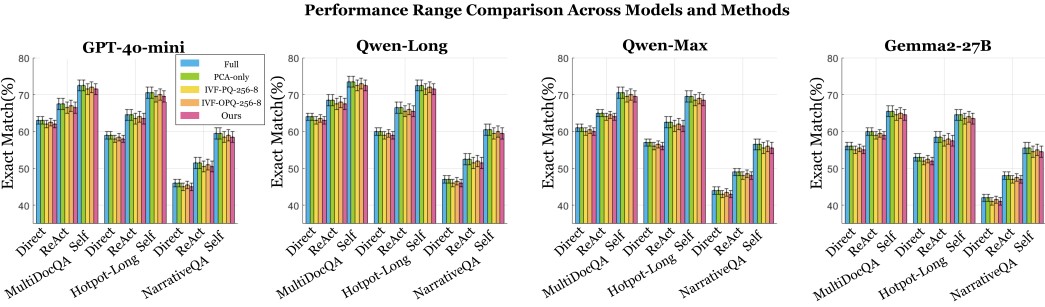

Figure 5: Performance across reasoning paradigms (Direct, ReAct, Self-Act) on three QA benchmarks with four LLMs.

**Generalizability of SCMF Across Reasoning Paradigms.** We evaluate SCMF under three reasoning paradigms—Direct, ReAct, and Self-Act—to assess its applicability in downstream tasks. As shown in Figure 5, results on MultiDocQA, Hotpot-Long, and NarrativeQA indicate that SCMF performs on par with PCA-only, typically within one percentage point, confirming that semantic compression does not compromise reasoning effectiveness. Compared to the Full configuration, PCA-only generally matches Full, while SCMF remains very close, usually within one point. In contrast, IVF-PQ and IVF-OPQ consistently lag behind by 1–2 points across paradigms, reflecting the accuracy degradation from aggressive quantization. These findings highlight that although compression inevitably introduces some loss, SCMF preserves reasoning effectiveness far better than quantization-based methods while providing notable gains in storage efficiency and retrieval latency, thus demonstrating robustness and generality in supporting diverse reasoning pipelines.

## 5 CONCLUSION

This work investigates semantic compressed memory for retrieval-augmented generation and is motivated by two key limitations in existing RAG systems, the high storage cost of dense embeddings and the limited traceability of retrieved knowledge. We introduce SCMF, a memory-efficient extension of RAG with residual vector quantization, and employ a dedicated training objective and semantic memory bank that support efficient CRUD operations. We observe that the framework not only reduces retrieval latency and storage overhead but also preserves accuracy, achieving a new efficiency–accuracy balance among compressed retrieval methods.

## 6 REPRODUCIBILITY STATEMENT

We have made every effort in our paper to provide sufficient information ensuring good reproducibility of our work:

- **Methods and Implementation Details:** In Section 3 of the main text, we detail the design of SCMF, including document encoding, PCA projection, two-level RVQ compression, Semantic Memory Units (SMU) and Semantic Memory Bank (SMB), and retrieval/reranking processes. Appendix A.2 provides a notation table ensuring conceptual clarity.
- **Training and Inference Procedures:** Sections 3.3 and 3.4 present complete training and inference algorithm steps, covering PCA fitting, RVQ codebook learning, CRUD dynamic operations and other details, enabling step-by-step reproduction.
- **Experiments and Parameter Configurations:** Section 4.1 and Appendix A.3 list all experimental datasets (MultiDocQA, Hotpot-Long, NarrativeQA, GovReport, QASPER, STS-B) and fully present data splits, model input lengths, quantization parameters, retrieval candidate counts, context budgets, and inference constraints. Tables 5 and 6 summarize core parameters, experimental environments, and hardware/software dependencies ensuring strict reproducibility.
- **Evaluation Protocols:** Section 4 and the Appendix provide metrics used for retrieval and question answering tasks (EM, nDCG@10, MRR, etc.) and experiment reproducibility parameters, making result comparisons verifiable.

We believe the parameter tables and experimental configurations provided in the appendix are sufficient to support complete reproduction of SCMF. We are not releasing source code during review; if accepted, we will make implementation and scripts publicly available to facilitate community use and extension.

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

CONTENTS

## A APPENDIX

### A.1 ACKNOWLEDGE

This article used large language models (such as ChatGPT) as an auxiliary tool in the language polishing process, but did not use them in research conception and academic content generation

### A.2 NOTATION

All symbols used in this paper, including both the main text and appendix, are shown in Table 5.

Table 5: Notation used throughout the paper.

| Symbol | Meaning |
| --- | --- |
| DocEncoder | Fixed text embedding model that maps a Raw Knowledge Unit (RKU) or query to a high-dimensional vector. |
| RKU | Raw Knowledge Unit; atomic text segment (typically a paragraph/document chunk of $\sim$600 tokens). |
| $\mathbf{e} \in \mathbb{R}^d$ | High-dimensional embedding of an RKU or query produced by DocEncoder. |
| $d$ | Embedding dimensionality of $\mathbf{e}$; set to 3072. |
| $W \in \mathbb{R}^{d \times d_\ell}$, $\mu \in \mathbb{R}^d$ | PCA projection matrix and mean vector. |
| $\mathbf{z} \in \mathbb{R}^{d_\ell}$ | Latent vector after PCA: $\mathbf{z} = W^\top(\mathbf{e} - \mu)$. |
| $d_\ell$ | Latent dimensionality after PCA; set to 270. |
| $\mathcal{C}_1, \mathcal{C}_2$ | Codebooks (level-1 / level-2) used in two-level Residual Vector Quantization (RVQ). |
| $M$ | Codebook size (prototypes per level); set to 32k. |
| $L$ | Number of RVQ levels; set to 2. |
| $\mathbf{c}_{1,i} \in \mathcal{C}_1, \mathbf{c}_{2,j} \in \mathcal{C}_2$ | Selected codewords at level-1 and level-2 for indices $i$ and $j$. |
| $(i, j)$ | Discrete RVQ indices (level-1, level-2) identifying an SMU. |
| $\hat{\mathbf{z}}$ | SMU prototype (quantized latent): $\hat{\mathbf{z}} = \mathbf{c}_{1,i} + \mathbf{c}_{2,j}$. |
| SMU | Semantic Memory Unit; a discrete prototype indexed by $(i, j)$ representing one or more RKUs. |
| SMB | Semantic Memory Bank; repository storing SMUs and explicit links between SMUs and RKUs. |
| $\mathcal{P} = \{\hat{\mathbf{z}}_{i,j}\}$ | Set of all SMU prototypes stored in the SMB (the "SMU space"). |
| $\mathbf{z}_q$ | Query latent vector after PCA: $\mathbf{z}_q = W^\top(\mathbf{e}_q - \mu)$. |
| $K$ | Number of top SMUs returned by ANN search (Top-$K$); set to 16. |
| $B$ | Token budget for context assembly (query + evidence); set to 512. |
| $E_{\mathrm{ret}}$ | Retrieved evidence set assembled from SMU-linked RKUs. |
| $\mathcal{L}_{\mathrm{vq}}$ | Quantization distortion loss: $\|\mathbf{z} - \hat{\mathbf{z}}\|_2^2$. |
| $\beta$ | Weight for commitment-style term in the codebook objective; $\beta = 0.5$. |
| $\mathrm{sg}[\cdot]$ | Stop-gradient operator used in the commitment-style term. |
| ANN | Approximate Nearest Neighbor search over $\mathcal{P}$ using cosine similarity. |
| BM25 | Sparse retrieval method used to generate candidates per SMU. |
| $\Delta\cos$ | Absolute change in cosine distance before vs. after quantization. |
| MSE | Mean squared error between $\mathbf{z}$ and $\hat{\mathbf{z}}$. |
| Spearman $\rho$ | Spearman rank correlation (e.g., ranking consistency vs. PCA baseline). |
| J@20 | Jaccard overlap of top-20 nearest neighbors before vs. after quantization. |
| EM, nDCG@10, MRR | Retrieval/QA evaluation metrics (Exact Match, normalized DCG at 10, Mean Reciprocal Rank). |
| Top-$K$, Top-20 | Truncation levels for candidate selection (e.g., Top-$K$ SMUs; BM25 Top-20). |

## A.3 DETAILED TRAINING SETTINGS

The specific parameters used in this paper are shown in Table 6, while the hardware specifications are shown in Table 7.

Table 6: Core parameter configuration of SCMF.

| Category | Parameter | Value | Description |
|---|---|---|---|
| Input Representation | RKU Length | 600 tokens | Length of a raw knowledge unit (paragraph or document chunk). |
| | DocEncoder Dimension | 3072 | Dimensionality of high-dimensional embeddings generated by the fixed text encoder. |
| Semantic Compression | PCA Latent Dim. $(d_\ell)$ | 270 | Latent dimensionality after projecting the 3072-d embedding via PCA. |
| | RVQ Levels $(L)$ | 2 | Number of residual vector quantization (RVQ) stages. |
| | Codebook Size per Level $(M)$ | 32,768 (32k) | Number of prototypes per codebook. |
| | Total Codebook Size | 65,536 (64k) | Total number of unique prototypes from two-level codebooks ($M \times L$). |
| | Total Bits | 30 bits | Bits required to index an SMU ($\log_2(32k) \times 2 = 15 \times 2$). |
| Training Objective | Codebook Weight $(\beta)$ | 0.5 | Weight of the commitment loss term in the objective $L_{\mathrm{cb}}$. |
| Inference Retrieval | Top-K SMUs | 16 | Number of semantic memory units retrieved by approximate nearest neighbor search. |
| | BM25 Candidates | 20 | Number of raw sentence candidates retrieved per SMU using BM25. |
| | Dense Re-rank | 2 | Number of final evidence sentences per SMU after dense re-ranking. |
| | Summary Length | 25 tokens (EN), 30–50 chars (ZH) | Maximum length of generated summaries in decompression. |
| | Context Budget | 512 tokens | Maximum context length (query + evidence) fed into the LLM. |
| Memory Management | Deduplication Threshold | 0.8–0.85 | Cosine similarity threshold for detecting duplicate SMUs. |

Table 7: Training and experimental environment configuration.

| Category | Configuration |
|---|---|
| Training Hardware (Codebook) | GPU: RTX 4090 (24GB) 
 CPU: Intel Xeon Gold 
 Memory: 128 GB RAM 
 High-speed SSD for large-scale retrieval corpus and embeddings. |
| Runtime Hardware | GPU: $1\times$ RTX 4090 
 CPU: 8+ cores 
 Memory: 32+ GB RAM |
| Software Environment | Python: 3.9+ 
 Deep learning framework: PyTorch 2.1 
 Key libraries: FAISS (ANN search), Scikit-learn (PCA), Transformers (DocEncoder and LLMs) |
| Training Hyperparameters | Optimizer: AdamW 
 Learning rate: $1 \times 10^{-3}$ 
 Batch size: 512 
 Training steps/epochs: 20k |

## A.4    ALGORITHMS

The section presents two essential algorithms that form the foundation of the SCMF framework. Algorithm 1 outlines the training pipeline that efficiently constructs and populates a compressed Semantic Memory Bank (SMB) by transforming document embeddings through PCA dimensionality reduction and residual vector quantization. Algorithm 2 details the inference pipeline that leverages this compressed memory structure to retrieve relevant evidence for a given query within specified token budgets, ultimately providing the language model with contextually appropriate information for generating responses.

---

**Algorithm 1** Training pipeline of SCMF

---

**Require:** retrieval corpus $\mathcal{U}$; DocEncoder; PCA $(W, \mu)$; RVQ codebooks $\mathcal{C}_1, \mathcal{C}_2$; SMB
**Ensure:** Trained $\mathcal{C}_1, \mathcal{C}_2$ and populated SMB
    **Stage A: Unsupervised Pretraining (Codebook/SMB Construction)**
 1: Fit PCA on embeddings from $\mathcal{U}$ to obtain $(W, \mu)$
 2: **for** $x \in \mathcal{U}$ **do**
 3:     $\mathbf{e} \leftarrow \text{DocEncoder}(x)$
 4:     $\mathbf{z} \leftarrow W^\top(\mathbf{e} - \mu)$
 5:     $(i, j), \hat{\mathbf{z}} \leftarrow \text{RVQ}(\mathbf{z}; \mathcal{C}_1, \mathcal{C}_2)$
 6:     Update $\mathcal{C}_1, \mathcal{C}_2$ via residual k-means / EMA to reduce $\|\mathbf{z} - \hat{\mathbf{z}}\|_2^2$
 7:     Insert $\big(x \rightarrow (i, j), (i, j) \rightarrow \hat{\mathbf{z}}\big)$ into SMB
 8: **end for**

---

**Algorithm 2** Inference pipeline of SCMF

---

**Require:** Input query $q$; PCA $(W, \mu)$; token budget $B$; codebooks $\mathcal{C}_1, \mathcal{C}_2$; SMU prototype set $\mathcal{P}$
**Ensure:** Output generated by LLM based on retrieved evidence
 1: $\text{ctx} \leftarrow \text{RETRIEVEEVIDENCE}(q, (W, \mu), B)$; **return** $\text{LLM}(q, \text{ctx})$
    **Function** RetrieveEvidence$(q, (W, \mu), B)$
 2: $\mathbf{e}_q \leftarrow \text{DocEncoder}(q)$; $\mathbf{z}_q \leftarrow W^\top(\mathbf{e}_q - \mu)$
 3: $\text{SMUs} \leftarrow \text{ANN\_over}(\mathcal{P}, \mathbf{z}_q)$                     ▷ Retrieve Top-$K$ SMUs
 4: $E_{\text{ret}} \leftarrow \bigcup_{\text{SMU}} \text{Top-2}\big(\text{DenseRerank}(\text{BM25}(\text{Docs}(\text{SMU}), 20))\big)$
 5: **return** $\text{ASSEMBLE}(E_{\text{ret}}, B)$

---

A.5   STORAGE COST ANALYSIS

Table 8 shows that the storage costs of Full-Dense and PCA-only methods increase linearly with document scale, reaching tens of GiBs at very large scales. While PQ/OPQ and their IVF variants show some reduction, they still expand rapidly with document count. In contrast, SCMF's overhead primarily consists of a one-time codebook (approximately 67.5 MiB) and a minimal increment (3.75 B/document), maintaining a scale of only hundreds of MiB even with tens or hundreds of millions of documents, far lower than all baselines. This demonstrates SCMF's ability to effectively transform storage overhead from linear to near-constant growth, highlighting its scalability in large-scale retrieval scenarios.

Table 8: Extended Experiment: Storage Cost Growth with Document Scale

| Documents | Full-Dense | PCA-only (d=270) | PQ/OPQ (codes only, 8 B) | IVF-PQ/OPQ (codes+id≈16 B) | SCMF (67.50 MiB + 3.75 B/doc) |
|---|---|---|---|---|---|
| 1,000 | 11.72 MiB | 1.03 MiB | 0.0076 MiB | 0.0153 MiB | 67.50 MiB |
| 10,000 | 117.19 MiB | 10.30 MiB | 0.076 MiB | 0.153 MiB | 67.54 MiB |
| 100,000 | 1.14 GiB | 103.00 MiB | 0.763 MiB | 1.49 MiB | 67.86 MiB |
| 1,000,000 | 11.44 GiB | 1.01 GiB | 7.63 MiB | 15.26 MiB | 71.08 MiB |
| 10,000,000 | 114.44 GiB | 10.10 GiB | 76.29 MiB | 152.59 MiB | 103.26 MiB |
| 16,000,000 | — | — | 122.07 MiB | 244.14 MiB | 124.72 MiB |
| 20,000,000 | — | — | 152.59 MiB | 305.18 MiB | 139.03 MiB |
| 50,000,000 | — | — | 381.47 MiB | 762.94 MiB | 246.31 MiB |
| 100,000,000 | — | — | 762.94 MiB | 1.49 GiB | 425.13 MiB |

A.6   COMPLEXITY ANALYSIS

Table 9 summarizes the complexity differences of various methods during construction and maintenance phases. Full-Dense and PCA-only approaches offer simplicity for insertion and deletion operations, but their overall storage costs and one-time training/projection overhead increase rapidly with scale. While PQ/OPQ and IVF-PQ/OPQ methods can compress storage, they require additional codebook training or clustering operations, making incremental maintenance relatively complex. The IVF series methods particularly struggle with deletions, often requiring lazy reclamation that compromises efficiency. In contrast, SCMF only needs to complete PCA and RVQ training once, after which both insertion and deletion operations maintain constant complexity ($O(1)$). Updates can be efficiently completed through semantic inverted mapping. This design significantly simplifies the dynamic maintenance process while maintaining compressibility, demonstrating SCMF's advantages in large-scale, continuously evolving scenarios.

Table 9: Comparison of different vector indexing methods

| Method | Construction | Insert | Delete |
|---|---|---|---|
| Full-Dense | $O(Nd)$ write | $O(1)$ append | $O(1)$ mark |
| PCA-only | $O(Nd^2)$ SVD, $O(Nd\ell)$ proj | $O(d\ell)$ | $O(1)$ mark |
| PQ/OPQ | $O(Nd)$ train+encode | $O(d)$ | $O(1)$ mark |
| IVF-PQ/OPQ | $O(Ndk)$ clust, $O(Nd)$ enc | $O(\log k)$ | Complex |
| **SCMF** | $O(Nd^2)$ PCA, $O(Nd\ell)$ RVQ | $O(1)$ | $O(1)$ recycle |

