# OpenReview forum: "SCMF: Lightweight Retrieval-Augmented Generation via Retrieval Vector Compression"
_ICLR.cc/2026/Conference — ICLR 2026 Conference Withdrawn Submission_

### Official Review · Reviewer_Q9LQ · 2025-10-31

**Soundness:** 2
**Presentation:** 2
**Contribution:** 2
**Rating:** 2
**Confidence:** 4

**Summary:**

This paper presents SCMF (Semantic Compressed Memory Framework), a novel indexing and retrieval framework designed to improve the efficiency and scalability of Retrieval-Augmented Generation (RAG) systems. The key contribution lies in introducing a semantic compression mechanism that projects document embeddings into a low-dimensional semantic space using PCA, followed by Residual Vector Quantization (RVQ) to form compact Semantic Memory Units (SMUs). These units are linked to raw knowledge entries through a semantic inverted index, which enables efficient CRUD operations and preserves traceability.

**Strengths:**

1. This paper tackles an important and practical efficiency bottleneck in modern RAG systems.

2. This paper has presented experimental evaluations on QA benchmarks demonstrate that the proposed SCMF significantly lowers storage and retrieval latency,

**Weaknesses:**

1. On the technical core and novelty
If I understand correctly, this paper proposes to first apply PCA to project document embeddings into a lower-dimensional space, and then apply a two-level residual vector quantization (RVQ) to compress the projected embeddings. While the proposed pipeline is technically reasonable, it mainly combines existing techniques — PCA for dimensionality reduction and residual quantization for compression — both of which are well-established. Thus, the technical novelty appears limited. The main contribution seems to lie in terminology and framing rather than in algorithmic innovation.

Moreover, the paper introduces several new terms (e.g., Raw Knowledge Unit for a paragraph or document, Semantic Compressor for PCA, and Semantic Memory Unit for the quantized representation). However, the techniques behind these components are not inherently semantic in nature. PCA and RVQ are purely geometric and statistical transformations that do not model semantics explicitly. Therefore, using the term semantic throughout the paper may be misleading, as it overstates the conceptual contribution and creates a mismatch between terminology and method.

2. The claimed support for efficient CRUD (Create, Read, Update, Delete) operations is also not unique to this framework. CRUD operations are standard in most quantization-based or vector database systems that maintain indexable and updatable codebooks.


3. On experimental setup and fairness of comparison
The experimental comparison with PQ and OPQ appears not directly comparable. Specifically, the proposed SCMF uses two levels of RQ with 32K codebook size, while PQ and OPQ are configured with 8 codebooks and codebook size 256. These settings differ substantially in both codebook granularity and total representational capacity, making the comparison unfair. The paper should control for comparable total codebook size or quantization bit budget to ensure a fair comparison of compression efficiency and retrieval accuracy.

5. Several statements in the paper are inaccurate or misleading:

(1) Line 132: The paper claims that “our system stores only the RVQ codebooks (and short codes), making index size largely independent of d.” This is not entirely correct, since the codebooks themselves are d-dimensional. As d increases, the storage required for codebooks also increases, making the total index size still dependent on d.

(2) Line 348: The phrase “in the subsequent end-to-end retrieval experiments” is unclear. It is not evident in what sense the proposed method constitutes an end-to-end retrieval framework, since the retrieval and compression components appear modular and not jointly optimized.

(3) Line 377: The statement that “SCMF’s index size stays nearly constant across corpus scales” is inaccurate. Each document is associated with a discrete short code, so the total index size should scale linearly with the corpus size. The apparent constancy may result from the large codebook size (32K), which dominates the overall storage, but the underlying scaling property remains linear with the number of documents.

**Questions:**

Please see above weaknesses.

---

### Official Review · Reviewer_4DSV · 2025-11-01

**Soundness:** 1
**Presentation:** 2
**Contribution:** 1
**Rating:** 0
**Confidence:** 5

**Summary:**

This work proposes a technique (SCMF) for information retrieval for Retrieval-Augmented Generation (RAG). It leverages a combination of dense and sparse rerieval. According to the provided space complexity analysis (i.e., storage cost), the technique offers significant savings with respect to existing techniques.

**Strengths:**

- The technique offers competitive accuracy results compared to standard alternatives.
- A combination of using a variant of an inverted index based on vector quantization, BM25, and reranking with full-precision vectors is used to provide the desired results.

**Weaknesses:**

- The analysis of the related work severely lacks in depth and omits many solutions that are very related to the one proposed in this work. For example, the Inverted Multi-Index (IMI) proposes a similar indexing technique based on product quantization (PQ) instead of residual quantization. Besides acknowledging IMI in the related work, the authors should show why residual quantization (RQ) offers a better alternative when compared to RQ. Particularly, because PQ is easier and faster to train than RQ. Another very related work is FLANN, that builds a tree of residual quantizers to index vectors. Although FLANN does not leverage BM25 and relies on vectors alone, it should be mentioned and acknowledged. Moreover, graph indices are the most commonly used techniques for RAG retrieval in industrial deployments, see HNSW and DiskANN are standard  solutions at this point. Comparisons to all of these solutions should be included in the experimental section, as graph indeices are often considered state-of-the-art vector indices.

Babenko, Artem, and Victor Lempitsky. "The inverted multi-index." IEEE transactions on pattern analysis and machine intelligence 37.6 (2014): 1247-1260.

Muja, Marius, and David Lowe. "Flann-fast library for approximate nearest neighbors user manual." Computer Science Department, University of British Columbia, Vancouver, BC, Canada 5.6 (2009).

Malkov, Yu A., and Dmitry A. Yashunin. "Efficient and robust approximate nearest neighbor search using hierarchical navigable small world graphs." IEEE transactions on pattern analysis and machine intelligence 42.4 (2018): 824-836.

Jayaram Subramanya, Suhas, et al. "Diskann: Fast accurate billion-point nearest neighbor search on a single node." Advances in neural information processing Systems 32 (2019).

- More importantly, the authors claim that existing techniques "lack backtracking to the original knowledge." This is just not true. Any sensible deployment of a retrieval pipeline for RAGs (or other applications for that matter) uses full-precision vectors for reranking if the index relies on vector quantization. For example, DiskANN uses PQ for the initial search and full-precision vectors for reranking. Similar approaches are commonly used with PQ-based inverted indices. In practical deployments, the outputs of these indices are combined with BM25 to enhance the dense retrieval with term-based retrieval. This is just common standard practice today.

- The spatial analysis of SCMF does not seem accurate to me. SCMF is intrinsically using BM25 and full-precision vectors for reranking. As such, the footprint of both of these techniques need to be accounted for when evaluating that of SCMF. Apparently, the authors are just comparing one component of SCMF (the inverted index) that accounts for a tiny fraction of the total footprint with the entire footprint of techniques like IVF-PQ. The footprint of BM25 and full-precision vectors in SCMF dominates that of the inverted index. When considered its larger footprint, SCMF stops being lightweight and does not seem to lead to significant accuracy improvements over PQ or OPQ (even more so because if paired with full-precision vectors for reranking, thse techniques would show improved accuracy).

- In my opinion, SCMF does not truly have incremental indexing capabilities. Being a variant of an inverted index, SCMF will suffer significantly when facing distribution shifts. Complex operations are often needed to handle such cases, as discussed in DEDRIFT. Distribution shifts are the crux of incremental indexing, not the ability to insert or remove vectors (virtually all existing techniques provide this capability).

Dmitry Baranchuk, Matthijs Douze, Yash Upadhyay, and I Zeki Yalniz. 2023. DEDRIFT: Robust Similarity Search under Content Drift. In IEEE International Conference on Computer Vision. 11026–11035.

**Questions:**

- I recommend an in-depth literature analysis and study of standard retrieval pipelines for RAG, beyond the succinct sample that I provide in this review.
- The authors should justify why their spatial analysis is correct.
- The should justify why the clustering-based technique does not need updating and re-indexing when facing incremental indexing.

---

### Official Review · Reviewer_Azzi · 2025-11-01

**Soundness:** 2
**Presentation:** 2
**Contribution:** 2
**Rating:** 4
**Confidence:** 4

**Summary:**

This paper proposes a vector storage optimization approach for vector-based retrieval systems, aiming to reduce storage costs and retrieval latency through semantic compression and hybrid re-ranking. However, it has several major limitations, including a lack of comparison with closely related techniques, unclear framing of its main contribution, insufficient experimental evaluation, and no discussion of the impact of training on the proposed method.

**Strengths:**

- The paper proposes a new vector storage optimization approach which addresses storage cost and retrieval latency bottlenecks associated with large-scale corpora.
- The paper combines efficient ANN search in SMU space with hybrid re-ranking (BM25 sparse + dense retrieval) to achieve controllable storage costs and reduced latency while maintaining accuracy.

**Weaknesses:**

- The paper lacks comparisons with state-of-the-art ANN systems like DiskANN and SPTAG, which are specifically designed for large-scale vector search. Without benchmarking against these established systems, it's difficult to assess whether SCMF's claimed advantages in storage and latency are truly competitive or simply comparable to older methods (IVF-Flat, HNSW). This gap undermines the claim that SCMF represents a significant advancement in vector indexing.
- The paper is framed as a lightweight RAG method but only focuses retrieval performance. This creates confusion about the actual contribution. If SCMF is primarily a retrieval index optimization technique, which the paper suggests, it should be positioned as such rather than as a complete RAG solution.
- The datasets used, such as MultiDocQA, HotpotQA-Long, NarrativeQA, involve relatively small-scale corpora that don't reflect real-world deployment scenarios.
- The paper provides unclear training specifications. For example, what data was used to train the VQ-VAE-inspired compression model, how much training data is needed for effective SMU learning, what is the impact of training data domain/size on retrieval performance,
can the learned compression generalize to out-of-domain documents?

**Questions:**

Please see my comments provided above.

---

### Official Review · Reviewer_2SvU · 2025-11-03

**Soundness:** 2
**Presentation:** 2
**Contribution:** 1
**Rating:** 2
**Confidence:** 3

**Summary:**

The paper presents a method for ann search using quantization. It uses PCA and a two-level codebook learned using kmeans on the data. The experiments show the method outperforms classical vector quantized approaches

**Strengths:**

The experiments show positive results

**Weaknesses:**

- The paper lack novelty. The main contribution of the paper is to use pca + kmeans for vector quantization. That's a standard technique

- The paper does not engage with the vector quantization literature, nor uses any recent baselines. There are many work on vector quantization for retrieval, (https://arxiv.org/pdf/2501.03078, https://arxiv.org/pdf/2405.12497, https://arxiv.org/pdf/2304.04759) and the references therein, and they should be included as baselines

- The paper should present dataset specific results in table 3 instead of aggregate results

**Questions:**

NA

---

### Note · Authors · 2025-11-15

I have read and agree with the venue's withdrawal policy on behalf of myself and my co-authors.